# IL-12 and IL-23—Close Relatives with Structural Homologies but Distinct Immunological Functions

**DOI:** 10.3390/cells9102184

**Published:** 2020-09-28

**Authors:** Doreen M. Floss, Jens M. Moll, Jürgen Scheller

**Affiliations:** Institute of Biochemistry and Molecular Biology II, Medical Faculty, Heinrich-Heine-University, D-40225 Düsseldorf, Germany; jens.moll@uni-duesseldorf.de (J.M.M.); jscheller@uni-duesseldorf.de (J.S.)

**Keywords:** IL-12 family cytokines, IL-12, IL-23, signaling, autoimmune diseases

## Abstract

Cytokines of the IL-12 family show structural similarities but have distinct functions in the immune system. Prominent members of this cytokine family are the pro-inflammatory cytokines IL-12 and IL-23. These two cytokines share cytokine subunits and receptor chains but have different functions in autoimmune diseases, cancer and infections. Accordingly, structural knowledge about receptor complex formation is essential for the development of new therapeutic strategies preventing and/or inhibiting cytokine:receptor interaction. In addition, intracellular signaling cascades can be targeted to inhibit cytokine-mediated effects. Single nucleotide polymorphisms can lead to alteration in the amino acid sequence and thereby influencing protein functions or protein–protein interactions. To understand the biology of IL-12 and IL-23 and to establish efficient targeting strategies structural knowledge about cytokines and respective receptors is crucial. A highly efficient therapy might be a combination of different drugs targeting extracellular cytokine:receptor assembly and intracellular signaling pathways.

## 1. Introduction

Cytokines are important mediators of the immune system. They are involved in pathophysiological processes including autoimmunity and cancer development. In contrast to hormones, cytokines are produced by more than one cell type. Most cytokines act locally in an autocrine or paracrine fashion. Exceptions are cytokines, which enter the bloodstream and signal endocrine, or membrane-bound cytokines, which act juxtacrine. Cytokines are classified according to their function or structure, or based on the receptors they bind. The latter is a reasonable classification because cytokines bind to multimeric receptor complexes in which often one subunit is also found in receptor complexes utilized by other cytokines [1,2]. Accordingly, cytokines and cytokine receptors are subdivided into seven major families [3]. The largest family are class I cytokine receptors and corresponding cytokines. A central feature of class I cytokines is a common up-up-down-down topology of their four antiparallel α helices. Therefore, these cytokines have also been referred to as the α-helical bundle cytokine family [4]. Class I cytokines are further subdivided based on the shared receptor subunit [3]. One group are IL-6 family cytokines, which are defined as cytokines that use the common signaling receptor subunit gp130 and originally comprise eight cytokines—IL-6, IL-11, IL-27, oncostatin M (OSM), leukemia inhibitory factor (LIF), ciliary neurotrophic factor (CNTF), cardiotrophin 1 (CT-1), and cardiotrophin-like cytokine factor 1 (CLCF1) [2]. IL-35, and IL-39 have been recently added to the family because they share gp130 in their receptor complexes [5]. A second group of class I cytokines are IL-12 family cytokines. Members of the IL-12 family form soluble heterodimers consisting of a four-helix-bundle α subunit (IL-23p19, IL-27p28 or IL-12p35) and a β subunit (Epstein–Barr virus-induced gene 3, EBI3 or IL-12p40), which is structurally related to soluble IL-6 receptor alpha (sIL-6Rα) [6]. At the moment, the IL-12 cytokine family has five members—IL-12, IL-23, IL-27, IL-35 and IL-39 (Table 1, Figure 1). IL-27, IL-35 and IL-39 are shared with the IL-6 family. A possible sixth member, called IL-Y, has currently only been described as a genetically engineered heterodimer IL-27p28/IL-12p40 [7,8].

Homodimeric members have also been described for IL-12p40 (p80) [21] and IL-27p28 (IL-30) [22]. Surprisingly, the IL-12 family is still growing because new combinations forming heterodimeric cytokines containing at least one α or β subunit have been identified (IL-12p40/CD5L [23], IL-12p35/CRLF1 [24], IL-6/EBI3 [25], IL-27p28/CRLF1 [22], IL-27p28/sIL-6R [26]). Receptor complexes of IL-12 family cytokines are also heterodimeric and consist of IL-12Rβ1, IL-12Rβ2, IL-23R, WSX-1 and gp130 (Figure 1, [27]). IL-12 signals via IL-12Rβ1 and IL-12Rβ2, and IL-23 signals via IL-12Rβ1 and IL-23R, whereas IL-27 and IL-35 shares receptor subunits with the IL-6 family (gp130/WSX1 and gp130/IL-12Rβ2, gp130/gp130, IL-12Rβ2/IL-12Rβ2, WSX1/IL-12Rβ2, respectively) [6]. Using engineered chimeric cytokine receptors we showed, that additional combinations of receptors are possible [28]. Based on their shared use of gp130 IL-6 and IL-12 cytokines are often considered members of the IL-6/IL-12 family comprising both IL-6 and IL-12 family cytokines [29]. Functionally, IL-6/IL-12 cytokines are key regulators of the immune system. They have pleiotropic functions and play critical roles in multiple immune responses, and a variety of physiological and pathological processes [27].

Initially, IL-12 was thought to be the driving force in several autoimmune disease models, but this was challenged in 2003 by a study from Cua and colleagues [30]. They demonstrated that mice lacking IL-23 (p19^−/−^ and p40^−/−^ mice) were resistant to experimental autoimmune encephalomyelitis (EAE) in contrast to IL-12 deficient (p35^−/−^) mice [30]. This was consistent with studies from Zhang and colleagues. They found that IL-12 responsiveness via IL-12Rβ2 is not required in the induction of EAE [31]. Additionally, they showed that IL-12Rβ1^−/−^ mice, lacking IL-12 and IL-23 signal transduction, are completely resistant to EAE [32]. The pilot study from Cua and colleagues defined a specific role for IL-23 in autoimmune diseases [33]. Furthermore, this study was the starting point to identify mechanisms by which the IL-12 family contributes to immune mediated diseases [6]. Over the last 20 years, preclinical data and clinical trials identified a clear relationship between IL-12 and IL-23 in tissue inflammation and tumor growth [34]. Biological consequences of IL-12 and IL-23 signaling in various diseases and therapeutic targeting strategies have been described in various reviews [35,36,37,38,39,40,41]. Table 2 summarizes similarities and differences of both cytokines with regard to immune responses and diseases. Effective targeting of IL-12 and IL-23 pathways in animal models and clinical trials broadened clinical indications of IL-12/IL-23 effectors in immune-mediated diseases and cancer [42]. This review focuses on structural features of IL-12 and IL-23, cytokine:receptor complex formation, signal transduction and functional consequences of single nucleotide polymorphisms (SNPs) within IL-23R. Structural knowledge about cytokine:receptor interaction is indispensable for the establishment of therapeutic strategies.

## 2. Structural Features of IL-12, IL-23 and Their Receptors

IL-12 cytokines function as heterodimers comprising two subunits, α and β (Table 1). Similarly to members of the IL-6 family, the α subunits IL-12p35 and IL-23p19 of IL-12 and IL-23 contact their shared β subunit IL-12p40 through residues located in helices A and D and the AB loop connecting helices A and B. The corresponding residues of IL-12p40 are located in domains D2 and D3. The β subunits are linked to the α subunits through disulfide bonds formed between C96 of IL-12p35 and C73 of IL-23p19 and C199 of the IL-12p40 subunit for human IL-12 and IL-23, respectively (Figure 2). Amino acids are numbered according to the database entry starting with the signal peptide. However, IL-27 and IL-35 are not secreted as disulfide-linked heterodimers [49]. IL-23 is a composite cytokine consisting of IL-23p19 and IL-12p40, whereas IL-12 is formed by IL-12p35 and IL-12p40 [11,13,50]. The latter was independently identified by two groups and named natural killer cell stimulatory factor (NKSF) and cytotoxic lymphocyte maturation factor (CLMF), respectively [10,12]. Secretion of bioactive IL-12 requires co-expression and disulfide bond formation of IL-12p35 and IL-12p40 in the cell [11,13,51]. IL-12p40 is found as free monomer or disulfide-linked homodimer and promotes stabilization and export of IL-12p35 [52,53]. IL-12p35 alone could not be detected [54]. Co-expression of its β subunit IL-12p40 inhibits IL-12p35 misfolding and allows secretion of biologically active heterodimeric IL-12 [55]. The intermolecular disulfide bridge is dispensable for formation and activity of IL-12 [51,56]. However, decreased IL-12 stability and secretion was reported in the absence of the inter-chain disulfide bond [57]. Nevertheless, one disulfide bond within IL-12p35 is critical for IL-12 secretion, stability and biological activity [55]. The disulfide-linked homodimer (IL-12p80) is secreted in excess compared to IL-12 [58]. Secreted IL-12p40 monomer and homodimers function as IL-12 antagonists [59,60]. IL-12p40 homodimers induced preventive and therapeutic effects in autoimmune arthritis models [61]. Secreted IL-12p40 also impaired IL-23-mediated responses [62]. However, a few studies reported IL-12p40 homodimer as an agonist in macrophage chemotaxis [63,64], an inducer of TNF and LTα in microglia and macrophages [65,66], and of iNOS in microglia [67].

Co-expression of IL-12p40 led to enhanced secretion of IL-23p19, a four-helix-bundle protein identified by a computational screen [9]. Oppmann and colleagues termed this novel composite cytokine IL-23. Whereas human IL-23p19 is not secreted, a small amount of murine IL-23p19 was detected in cell culture supernatants of HEK293T and COS-7 cells [9,68]. IL-23p19 and IL-12p35 subunits overlap, and their four-helix bundles match one another despite their low sequence homology of 15% (Figure 2D) [69]. Biologically active IL-23 requires secretion of IL-12p40 and IL-23p19 [9,68]. Both subunits are linked via an intersubunit disulfide bond formed at the top edge of the interface between human IL-23p19 C73 on the A-B loop to IL-12p40 C199 [70] (Figure 2B,C). The free cysteines C33, C41, and C73 of human IL-23p19 are bound by the PDI family member ERp44 resulting in proper ER quality control and assembly of IL-23 [71]. Bypassing ER quality control of IL-23p19 was achieved by stabilizing its first helix. The resulting optimized human IL-23p19 was independently secreted from mammalian cells [71]. Biologically active IL-23p19 was identified in endothelial cells, which was important for endothelial inflammation, leukocyte adhesion, and transendothelial migration [72].

Interactions of IL-12 type cytokines with their cognate receptors have been proposed based on the structure of IL-6, IL-6Rα and gp130 [73,74,75,76]. This “site 1-2-3” architectural model relies on early structural studies of the human growth hormone (hGH) and its receptor [77,78]. In the “site 1-2-3” structural model for IL-12 type cytokines site 1 is formed by the interface between the α and β cytokine subunits. The cytokine-binding homology region (CHR) of one receptor binds to the cytokine forming site 2, and the Ig domain of the second receptor interacts with the α subunit resulting in site 3 [74]. Site 1 mutations of IL-12p35 have been identified based on the crystal structure of IL-12 [51,57]. Mutation of R207 and Y211 in murine IL-12p35, R211 and Y215 in human IL-12p35, are critical for association with IL-12p40 [57] (Figure 2C). Importantly, the central arginine projects from IL-12p35 into a deep pocket on IL-12p40 with D312 at its base. This amino acid and additional residues forming this pocket (Y136, Y268, Y314 and Y315) are essential for IL-12 formation [51]. IL-12p40 uses the same binding surface to contact both IL-23p19 and IL-12p35 [70]. Targeted site 1 mutations in murine IL-23p19 and IL-12p40 prevented the formation of IL-23 [68]. We identified I176, A178, and R179 as critical hot spot amino acids of site 1 in murine IL-23p19 via reporter cellular assays in vitro [68]. Figure 2D displays the position of these residues in the human structure of IL-23. Furthermore, we introduced mutations Y265K and Y318K into domain 3 of murine IL-12p40 and interrupted binding to IL-23p19 [68]. Site 2 and site 3 are assembled of the cytokine subunits and respective receptors.

The receptors are type I membrane proteins and belong to the class I cytokine receptor family (hematopoietin family) which lack intrinsic kinase activity and depend on associated tyrosine kinases. The extracellular part of IL-12Rβ1 and IL-12Rβ2 consists of 5 fibronectin-type-III (FNIII) domains (Figure 1). Two of them formed the cytokine-binding homology region (CHR), which possess the binding regions for the cytokines. The IL-23R contains an unstructured stalk region instead of three membrane-proximal FNIII domains. This stalk region acts as a spacer to position the CHR at a defined distance from the plasma membrane to enable signal transduction [79]. Domains of the CHR are characterized by conserved cysteine residues for interstrand disulfide bonds and the WSXWS motif, which might be involved in receptor complex formation, ligand binding, signaling, receptor folding and export [80]. The cysteine residues and the WSXWS motif are also present in the β subunits of heterodimeric IL-12 type cytokines (Figure 1). IL-23R and IL-12Rβ2 contain amino-terminal Ig domains, which are required for site 3 interactions. IL-12 signals via the heterodimeric receptor consisting of IL-12Rβ1 and IL-12Rβ2. IL-12Rβ1 has homology to gp130 and was initially isolated from human tissue [81]. Afterwards, the murine homolog was cloned [82]. In 1996, IL-12Rβ2 was identified as second IL-12 receptor chain with similarities in sequence and structure to gp130. Human and murine IL-12Rβ2 chains contain tyrosine residues within their cytoplasmic regions and they are considered as signaling components of the receptor [83]. Additionally, IL-23 requires the unique IL-23R for signal transduction, which shows homology to gp130 and IL-12Rβ2 [50]. In 2018, hydrogen/deuterium exchange coupled to mass spectrometry (HDX-MS) demonstrated that IL-23 binds to the N-terminal Ig domain of IL-23R in solid state and also under more physiologically relevant conditions [84]. Hydrophobic residues within the D helices of IL-6 type cytokines are critical for the formation of a functional site 3 [74,85,86,87,88]. Based on the structure of human IL-23 W156 was predicted to be the hot spot amino acid for site 3 interaction with IL-23R (Figure 3) [70]. At the interface of human IL-23:IL-23R the N-terminus of the IL-23R is wedged between the AB loop and the D helix of IL-23p19. K164 of IL-23p19 forms a hydrogen bond to the I28 main chain. Surrounding the highly conserved W156, the interface is stabilized by mainly hydrophobic residues and an additional hydrogen bond formed between the main chain amide of W156 and D118 of the IL-23R. The AB loop of IL-23p19 contacts the D1 domain of the IL-23R via its N-terminus, I56, the G beta strand and residues H108 and Q110 of the loop connecting strands F and G (Figure 3).

In 2015, we expressed murine IL-23 with IL-23p19W157A and disrupted the interaction of the cytokine with murine IL-23R. Additionally, domains D2 and D3 of IL-12p40 were required and sufficient to facilitate the binding site 3 of IL-23 to IL-23R [68]. In 2018, the crystal structure of human IL-23:IL-23R in complex with a single-domain VHH camelid antibody targeting human IL-23 (Nb22E11) was solved and provided new structural insights into the formation of IL-12 type cytokine:receptor complexes (Figure 3) [89]. Accordingly, Bloch and colleagues proposed additional amino acids with importance for site 3 interactions and analyzed them in the murine system. They confirmed our data from Schröder et al. (2015) for murine IL-23p19W157A and showed no measurable interaction of the cytokine with IL-23R. In addition, they were not able to detect interaction of IL-23p19L161E with IL-23R by bio-layer interferometry, while in a cellular context IL-23p19L161E was not comparable with IL-23p19W157A and exhibited measurable bioactivity [89]. When we analyzed the hot spot amino acid W156 in human IL-23p19 in cellular assays, substitution to alanine did not affect signaling and proliferation of cells with hIL-23R and hIL-12Rβ1. In contrast, binding to hIL-23R was diminished [90]. However, combined substitution of W156A and L160E, which becomes buried at the complex interface, in human IL-23 resulted in an inactive cytokine variant on cells expressing human IL-23 receptors. Human IL-23 is also active on cells expressing murine IL-23 receptors and vice versa [50]. W156A substitution in human IL-23p19 was sufficient to prevent signaling of cells expressing murine IL-23 receptors, indicating differences in the binding interface of human and murine IL-23 receptor complexes [90]. The reason for this may be that the human IL-23R projects Y100 into the center of site 3 where it stacks against W156 and L160 of human p19. In the murine IL-23R this Y residue is replaced by H77, which is likely too far away from L161 to form additional hydrophobic contacts when the conserved W residue in murine IL-23p19 is mutated. This in turn leads to loss of activity of W157A mutants of murine IL-23p19.

This phenomenon was not observed by us when analyzing site 3 interactions of IL-12p35 and IL-12Rβ2 [90]. Sequence and structural alignments indicated that Y185 in murine IL-12p35 and Y189 in human IL-12p35 are critical for site 3 interactions with IL-12Rβ2 [70,74,89]. Indeed, single amino acid substitutions Y185R and Y189R prevented binding of IL-12 to murine and human receptors, respectively, and influenced signal transduction. Murine IL-12 is also active on human cells and accordingly, the IL-12 variant with Y185R substitution was biologically inactive [14,90]. For the IL-12:IL-12Rβ2 complex no crystal structure is available. Hence, we modeled the complex structure using the IL-23R as a template for the IL-12Rβ2. The resulting model and the structure of IL-12 in complex with ustekinumab (PDB 3MHX) was subsequently superpositioned onto the structure of the IL-23:IL-23R complex. Similarly to the IL-23 complex structure at site 3 a hydrophobic core is formed by Y189 of human p35 and G115, F99 and V100 of the D1 domain of the IL-12Rβ2. Y189 may also be able to form a hydrogen bond to Q112 of the IL-12Rβ2 (Figure 3E,F). A substitution of Y189 with an R residue may lead to loss of hydrophobic contacts at the center of site 3 formed by IL-12p35 Y189, F99, and V100 of IL-12Rβ2. In contrast to the interface formed by human IL-23p19 and the IL-23R where L160 contributes to the hydrophobic interactions such additional interactions are not formed by the corresponding IL-12p35 residue I193 and thus Y189R results in an inactive cytokine.

The shared IL-12Rβ1 is responsible for site 2 interactions of IL-12 and IL-23. Twenty years ago, Ling and colleagues suggested that IL-12p40 contains the essential epitopes for receptor binding [21]. The shared IL-12p40 subunit allows both cytokines to engage IL-12Rβ1, and binding of IL-23 to IL-12Rβ1 was reported by Oppmann and colleagues [9,91]. This direct interaction of IL-12p40 with IL-12Rβ1 results in the antagonistic properties of IL-12p40 [21,91]. In 2015, we show that domain D1 and D2 of murine IL-12p40 are sufficient for interaction with IL-12Rβ1, accordingly interaction of mIL-23 to mIL-12Rβ1 is independent of murine IL-23p19. Here, amino acids Y143, L116, V117 and S118 of murine IL-12Rβ1 served as critical residues for site 2 interaction [68]. Furthermore, isothermal titration calorimetry provided direct evidence, that the low-affinity interaction between IL-23 and IL-12Rβ1 is exclusively mediated by the IL-12p40 subunit [89]. Due to the fact, that IL-23 comprising IL-23p19 and IL-12p40D2D3 exclusively interacts with IL-23R and not IL-12Rβ1, site 2 hot spot amino acids might be in IL-12p40 domain D1 [68]. The crystal structure of human IL-23 and its receptor IL-23R revealed how IL-23R activates IL-23 for recruitment of IL-12Rβ1. Assembly of the IL-23 receptor signaling complex is mediated via sequential recruitment of IL-23R and IL-12Rβ1. Here, the primary interaction (site 3) is mediated via the Ig-like domain of IL-23R to IL-23p19, which loosely contact IL-12p40 via a secondary interaction site (site 1). This interaction partially restructured the IL-23p19 subunit of IL-23 and restraints the IL-12p40 subunit resulting in high affinity binding of IL-12Rβ1 (site 2) [89]. This mechanistic interaction paradigm segregates cognate and shared receptor binding to α and β subunits of the heterodimeric IL-12 cytokine family.

## 3. Insights into IL-12 and IL-23 Signal Transduction

The formation of the cytokine:receptor complex induces the activation of complex signaling cascades within the cell. Accordingly, cytokine receptor expression on cells is a prerequisite for signal transduction. Signaling of IL-12 type cytokines occurs via heterodimeric receptor complexes (Figure 1). IL-12Rβ1 and gp130 are generally considered to be constitutively expressed and variation in heterodimeric receptor expression appears to rest with the α receptors IL-23R, IL-12Rβ2 and WSX-1 [6]. Chognard and colleagues analyzed gene expression of *IL23R* and *IL12RB2* in mouse and human and showed a restricted expression to specific cell types. IL-12Rβ2 is expressed by NK cells and at low levels in γδT cells, whereas IL-23R expression is restricted to specific T cell subsets (highest expression in CD8, γδT cells), a small number of B cells and innate lymphoid cells [92]. Interestingly, γδT cells showed detectable expression levels of IL-12Rβ1, IL-12Rβ2 and IL-23R [92]. Antigen-presenting cells (APCs), which are present or recruited to barrier tissues, e.g., gut, skin and lung mainly produce IL-23 [93]. The cytokine promotes the maturation of pathogenic TH17 cells expressing IL-17, IFN-γ, IL-22 and GM-CSF and resulting in chronic tissue inflammation [6,94]. In general, two different TH17 subtypes have been described: pathogenic TH17 cells producing IL-17 and IFN-γ, and nonpathogenic TH17 cells secreting IL-17 and IL-10 [94]. Accordingly, TH17 cells are involved in immune responses to extracellular bacteria and fungi. Furthermore, they play a pivotal role in autoimmunity [95]. Together with IL-6 and IL-21, IL-23 induces the activation of STAT3, which is important for the induction of RORγT [96], the key transcription factor for the differentiation of TH17 cells [97]. IL-12 is primarily produced by pathogen-activated APCs, particularly macrophages and dendritic cells (DCs) [98]. IL-12 signal supports the upregulation of T-bet in CD4^+^ cells and promote their differentiation into TH1 cells secreting IFN-γ [6]. T-bet is the master transcription factor for TH1 cells, which are important for host defense against intracellular pathogens, e.g., viruses, protozoa and bacteria. In addition, they are also responsible for the development of certain forms of organ-specific autoimmunity [95]. 

Binding of the heterodimeric cytokines IL-12 and IL-23 induces receptor dimerization of IL-12Rβ1/IL-12Rβ2 and IL-12Rβ1/IL-23R, respectively. These receptors lack intrinsic kinase activity and depend on associated tyrosine kinases Jak2 and Tyk2 [50,99] (Figure 4). Four mammalian Janus kinases (Jak1, Jak2, Jak3, and Tyk2) have been identified as indispensable for cytokine signaling [100]. Remarkably, Tyk2 deficient animals showed complete resistance against EAE [101]. Additionally, Tyk2 is involved in several IL-12/TH1- and IL-23/TH17-mediated animal models [102]. Finally, a protective single nucleotide polymorphism (SNP) within Tyk2 was identified (Tyk2-P1104A) and diminished IL-12 and IL-23 signaling [103]. These aspects highlight the importance for Tyk2 in IL-12 and IL-23 signaling. Tyk2 associates with the shared IL-12Rβ1 and Jak2 with IL-12Rβ2 and IL-23R, respectively [104,105]. Box1 and Box2 motifs have been identified in the cytoplasmic tails of both IL-12 receptors [83,105] (Figure 4). Yamamoto and colleagues demonstrated that the cytoplasmic membrane proximal region of human IL-12Rβ2 is important for Jak2 binding [106]. The IL-23R did not contain Box1 and Box2 motifs, but we characterized an unusual Jak2 binding site by the use of deletion and site directed mutagenesis [105]. Jak2 deficient animals cannot be used to analyze IL-12 and IL-23 signaling because they are embryonic lethal [107]. 

Cytokines of the IL-6/IL-12 family activate the Jak/STAT pathway, the MAPK pathway and the PI3K/Akt pathway [6]. The Jak/STAT pathway is the most studied one and involves the rapid transmission of the signal resulting in transcription of cytokine-responsive genes. Suppressor of cytokine signaling (SOCS) proteins are induced by STATs and act as negative feedback inhibitors of the Jak/STAT pathway. The SOCS family consists of eight members. SOCS1 and SOCS3 possess a kinase inhibitory region and can inhibit Janus kinases [108]. Inhibition of IL-6 family cytokines by SOCS3 is well characterized and general principles can be transferred to other class I cytokine receptors [109]. Different cytokine receptors bind different Jaks, so they also bind different STATs [110]. Seven mammalian STATs are known (STAT1, STAT2, STAT3, STAT4, STAT5A, STAT5B and STAT6) which contribute to signaling of different cytokines [110]. STAT1, 3, 4, and 5 are activated by IL-12 and IL-23. However, STAT4 is the dominant signaling molecule for IL-12 and STAT3 for IL-23 [6]. Cytokines of the IL-6 family activate the MAPK pathway via phosphatase SHP2, which binds to phosphotyrosine residue within the cytoplasmic domain of the receptor [111]. Accordingly, a SHP2 interaction motif (human: Y397EDI, mouse: Y416EDI) has been proposed for IL-23R [50,112]. We could show that Erk1/2 was activated in Ba/F3 cells expressing murine IL-12Rβ1 and IL-12Rβ2 upon IL-12 stimulation [105]. Over the past years, new players in IL-12 and IL-23 signaling have been identified [113,114]. For IL-23, they were recently reviewed by Pastor-Fernández [115]. One example is the activation of NFκB, which was required for IL-23 induced expression of IL-17 [116].

Binding of IL-12 or IL-23 to their respective receptors induces receptor heterodimerization followed by transactivation of Jak2 and Tyk2. Activated Jaks phosphorylate tyrosine residues within the cytoplasmic tail of the receptors which serve as docking stations for signaling proteins (e.g., STATs, [110]). The shared human IL-12Rβ1 does not contain any tyrosine residue within the intracellular domain (ICD); however, one tyrosine with obviously no function in signaling is present in murine IL-12Rβ1 (Y635) [82] (Figure 4). Accordingly, IL-12Rβ1 was considered as ligand-binding receptor at least in mice [117]. Furthermore, we described IL-12Rβ1 as helper receptor, which contains Box1 and Box2 motifs for association of Tyk2 but no STAT recruitment sites (SRS). Mutation and deletion of Box1 within mIL-12Rβ1 prevented Tyk2 binding and signal transduction of IL-23 and IL-12 [105]. However, macrophage migration and induction of NOS via IL-12Rβ1 have been reported [63,64,67]. Murine IL-12Rβ2 contains 11 tyrosine residues (Y677, Y693, Y727, Y737, Y738, Y748, Y757, Y778, Y804, Y811, Y866, UniProt P97378) within the cytoplasmic domain. Three of them are conserved in human IL-12Rβ2 (Y678, Y767, Y800, UniProt Q99665, [83] (Figure 4). Tyrosine residues at positions 757, 804, and 811 of murine IL-12Rβ2 are capable of mediating STAT4 tyrosine phosphorylation, whereas 737, 804, and 811 are important for STAT3 phosphorylation [118]. In addition, Y811F mutation reduced IL-12-induced T cell proliferation and IFN-γ production [118]. The homologous tyrosine in humans is Y800, which is required for STAT4 activation [119]. Additionally, the leucine residue at pY800 +1 position is required for STAT4 binding [120]. We demonstrated the activation of STAT1 and STAT3 in Ba/F3 cells with murine IL-12 receptors upon IL-12 stimulation [105]. Ba/F3 cells have very low STAT4 protein levels and cannot be used for analysis of IL-12-induced STAT4 activation [104]. Tyrosine residue 800 in human IL-12Rβ2 also served as docking site for SOCS3. Binding of SOCS3 to hIL-12Rβ2 inhibited IL-12-induced activation of DNA binding and transcriptional activities of STAT4 [121]. Additionally, SOCS1 regulates the response of IL-12 in vivo [122]. However, SOCS3 deficiency is not a critical regulator of TH1 polarization [123]. IL-12Rβ2 homodimers are proposed to be receptors for IL-35 [76]. We could demonstrate the functionality of IL-12Rβ2 homodimers by the use of chimeric cytokine receptors and activation of STAT3 and Erk1/2 [28]. This supports the dominant role of IL-12Rβ2 in IL-12 signaling.

The IL-23R is considered as the signal transducing receptor [105]. Using synthetic cytokine receptors we were able to demonstrate that IL-23R homodimers are biologically active. Synthetic IL-23R homodimers induced Jak2, STAT3, Erk1/2 and Akt phosphorylation [124]. In addition, biologically active human and murine IL-23R homodimers have been generated by deletion of the receptor stalk region [79]. Accordingly, we speculate that also other receptors of the IL-12 cytokine family can be biologically active as homodimers [125]. Murine and human IL-23R proteins show 66% identity on the amino acid level and contain seven intracellular tyrosine residues, whereof 6 are conserved. It was postulated that IL-23 signaling is mediated via three tyrosine residues in mice (Y416, Y504, Y626) and humans (Y397, Y484, Y611) [50]. Using mutation and deletion variants of murine and human IL-23R we demonstrated that predicted STAT3 binding sites (murine: Y504 and Y626, human: Y484 and Y611) mediate STAT3 activation. In addition, Y542 of murine IL-23R, which is not conserved in human IL-23R, also acts as STAT3 binding site. Furthermore, we identified a non-canonical, phosphotyrosine-independent STAT3 activation motif within the cytoplasmic domain of murine IL-23R [112]. We and others showed that IL-23 induced sustained STAT3 activation despite increasing expression of SOCS3 indicating IL-23R might be no target of SOCS proteins [112,124,126]. However, IL-23-induced STAT3 activation was enhanced in the absence of SOCS3 in T cells and loss of SOCS3 resulted in enhanced TH17 cell differentiation and cytokine expression [123]. It was already demonstrated that SOCS3 promoted IL-17 expression by human T cells [127] and IL-23 induced SOCS3 in TH17 cells [128]. We could also show that the predicted SHP2 binding site Y416 in murine IL-23R and an additional C-terminal part of the receptor are needed for phosphorylation of Erk1/2. Additionally, Y416 might be important for activation of PI3K/Akt pathway [112]. Based on these data, we generated IL-23R signaling deficient mice (IL-23R-Y416FΔICD). However, when analyzing these mice we failed to reproduce the positive or negative effects of IL-23 on myocardial infarction [129]. The understanding of IL-12 and IL-23 signal transduction and the knowledge about activated signaling proteins offer new possibilities for therapeutic strategies to target IL-12- and/or IL-23-induced diseases.

## 4. Disease and Functional Consequences of Single Nucleotide Polymorphisms in the IL-23R

Single nucleotide polymorphisms (SNPs) or single nucleotide variations (SNVs) are causally linked to individual phenotypic characteristics, caused by changes in gene expression, protein stability, localization and function. Most SNPs are found in non-coding sequences, including promoters, enhancer, introns and their influence might affect mRNA expression, splicing and stability. The SNP database (dbSNP) is an archive for a broad collection of polymorphisms in which SNPs are categorized and marked with a locus accession number “reference SNP” (rs). For the human IL-23R, dbSNP lists 31,233 SNPs, including intronic variations (n = 30,410), in-frame deletions (n = 3), insertions (n = 1), initiator codon variants (n = 1), missense variants (n = 370, some include the same mutated bases, but different exchanges and different amino acid substitutions, total number of exchanges is n = 398), non-coding transcript variants (n = 546), and synonymous variants (n = 157). Since we focused on ligand–receptor interaction and signal transduction in this review, we surveyed those 404 mutations in the coding sequence with direct consequences for the amino acid sequence of the hIL-23R (Figure 5A). The highest ratio of these SNPs per amino acid count within a domain was found in the signal sequence of the IL-23R (0.95 SNPs/aa), followed by D3 (0.76 SNPs/aa), whereas the lowest number of SNPs per aa were observed in the stalk region (0.47 aa/SNPs). This was surprising to us, because the stalk region is considered to be mainly unstructured and should therefore tolerate the most amino acid exchanges, which means that this region should experience the lowest evolutionary pressure to maintain a defined amino acid sequence. On the other hand, the structured D3 has important functions for the binding of IL-23 and should succumb to a comparably higher evolutionary pressure and therefore, at least in theory, might tolerate less amino acid exchanges. We can, however, not exclude that most missense SNPs did not or only marginally influence the function/stability of the IL-23R. A comprehensive list of all non-synonymous SNPs and the resulting amino acid exchanges are summarized in Appendix A. We have indicated SNPs in domain 1 that might have an influence on IL-23R/IL-23 interaction. Some have already been suggested by Bloch and colleagues (internal SNP number (ISN) 24; rs1465641771; N27S; domain 1 – ISN 25; rs1162714374; I28K; domain 1 - ISN 68+69; rs1345386569; T102P/A; domain 1 - ISN 72; rs1368052402; K107I; domain 1 – ISN 73; rs757367045; H108Y; domain 1) [89] and additional SNPs were identified during our re-evaluation of the human IL-23R/IL-23p19 interface (Provean [130] and structure-guided analysis [131]) and illustrated in Figure 5B. In detail, we have used the crystal structure of the human IL-23:IL-23R complex (PDB 5MZV) to analyze whether SNPs are located within the interface formed by IL-23 and the IL-23R. Sidechains of SNPs within the site 3 interface and within 5 Å distance of IL-23p19 and IL-12p40 sidechains were identified using UCSF Chimera. These SNPs may have effects on IL-23:IL-23R complex formation and are displayed in stick representation.

Interestingly, among the 404 annotated SNPs, which have an influence on the protein sequence, the vast majority was not linked to any disease or simply just not studied until now (396 out of 404 SNPs), most likely because they were found in more random-like genome sequencing approaches and not by an underlying disease-driven sequencing strategy. Therefore, for most missense SNPs it is completely unclear, if they influence receptor activity in terms of stability, folding, transport, cytokine binding, and/or signal transduction. A combined approach using molecular modelling followed by in-depth biochemical analysis might result in the identification of functional SNPs and subsequent connection to diseases, in particular for very rare SNPs, which will likely not show up in classical GWAS. An alternative experimental approach to manage the expanding SNP landscape is to identify missense mutations with altered biological function followed by the subsequent analysis of disease association.

The eight missense SNPs that were described in peer-reviewed publications are assumed to be hypomorphs that dampen IL-23R signaling to confer mainly protective properties. A variant with increased activity was not described. Here, we have shortly summarized what is currently known about these eight SNPs in the IL-23R with respect to disease association and receptor activity.

### 4.1. Internal SNP Number 03; rs1884444; Q3H; 43 Publications

Q3H (rs1884444) is an amino acid exchange of a polar but uncharged amino acid into a positively charged amino acid residue in the signal peptide of IL-23R, which might influence the properties of signal peptide recognition or processing. Some disease associations were found including a decreased risk to develop schistosomiasis-associated immune reconstitution inflammatory syndrome [132], increased gastric cancer [133], and esophageal cancer susceptibility [134]. Rs188444 might also lower the risk for ulcerative colitis and psoriasis [135], but not for Crohn′s disease [136]. However, no functional analysis has been performed with this receptor variant.

### 4.2. Internal SNP Number 58; rs76575803; R86Q; 1 Publication

R86Q is located in the extracellular domain 1. A positively charged amino acid is exchanged by a polar uncharged amino acid residue. R86Q is a rare allele and protective against Crohn′s disease [137]. It was suggested that it change the binding properties of IL-23R to IL-23 [89], but functional data are lacking.

### 4.3. Internal SNP Number 93; rs76418789; G149R; 8 Publications

G149R (rs76418789) is located in the extracellular domain 2 of the IL-23R. A hydrophobic amino acid is replaced by a positively charged amino acid. Protection for colitis ulcerosa and Crohn′s disease was associated with this SNP [137,138,139]. The G149R IL-23R variant is retained in the endoplasmic reticulum as an unfolded polypeptide, thereby reducing cell surface levels and resulting in diminished cellular activation by IL-23 [140]).

### 4.4. Internal SNP Number 109; rs371531867; Y173H; 1 Publication

Y173 is located in the extracellular domain 2. The Y173H variant does not differ in receptor activity from the wild-type IL-23R [126]. A disease association has not been studied to date.

### 4.5. Internal SNP Number 111; rs11465797; T175N; 1 Publication

T175N (rs11465797) is also located in the extracellular domain 2. No disease association was found for acquired aplastic anemia [141] and no functional characterization was performed. 

### 4.6. Internal SNP Number 198; rs7530511; L310P; 39 Publications

L310P is located in extracellular domain 3. In 39 publications (dbSNP), the mutation may contribute to the development of intracerebral hemorrhage [142], has no effect on inflammatory bowel disease [143], but is protective against psoriasis [144] and results in increased susceptibility against Graves’ ophthalmopathy [145]. Functional characterization is still lacking.

### 4.7. Internal SNP Number 231; rs41313262; V362I; 4 Publications

V362I is located in the transmembrane domain. V362I was shown to be protective against Crohn’s disease and ulcerative colitis [137]. The V362I IL-23R variant showed reduced protein stability, resulting in reduced cell surface expression levels and diminished cellular activation by IL-23. The cell surface half-life of the receptor variant was reduced in HEK293 and HeLa cells [140]. Moreover, lymphoblastoid cell lines derived from individuals where these haplotypes were present showed reduced expression of the receptor at the basal level [140].

### 4.8. Internal SNP Number 241; rs11209026; R381Q; 210 Publications

R381Q is located between the putative Jak2 binding site and the transmembrane domain in the cytoplasmic region of IL-23R protein. R381Q confers protection against ulcerative colitis [143,146,147], Crohn´s disease [137,146], ankylosing spondylitis [148] and psoriasis [144], increased susceptibility to rheumatoid arthritis [149] and was not linked to Graves’ ophthalmopathy [145]. The R381Q IL-23R variant had reduced receptor expression and decreased STAT3 phosphorylation upon stimulation with IL-23 in primary T cells, indicating that this is also a loss-of-function allele [150]. This was supported by analysis in HEK293 and HeLa cells. Here, the R381Q variant showed reduced protein stability resulting in diminished cellular activation by IL-23 due to a reduced cell surface half-life [140]. Lymphoblastoid cell lines derived from individuals also showed reduced expression of this receptor variant at the basal level [140]. In contrast, others did not observe reduced receptor activation and signal transduction in retrovirally transduced human T cell blasts [126].

## 5. Conclusions and Perspectives for New Targeting Strategies

This review focuses on molecular and structural features of IL-12 type cytokines and their cognate receptors. These cytokines are important players in the immune system and represent attractive targets for effective therapies. Several studies showed that the IL-12 type cytokine family is highly dynamic and still growing. New members were identified but there are some additional options for new relatives. IL-12 and IL-23 are closely related cytokines, which share cytokine subunits and receptor chains. IL-12 was described almost 30 years ago and IL-23 10 years later. The corresponding receptor chains were discovered shortly thereafter. In recent years, detailed structural information was observed for the cytokine:receptor binding. In addition, numerous SNPs have been identified in the coding sequence of hIL-23R with direct consequences for the amino acid sequence. The functional characterization for most of the hIL-23R SNPs is still missing. There are still absent parts to complete the whole puzzle of receptor complex formation. Understanding of cytokine:receptor binding and the knowledge of cytokine signaling are important prerequisites for the development of effective targeting strategies. IL-12 or IL-23 signaling can be simultaneously blocked by ustekinumab, a monoclonal antibody targeting the shared subunit IL-12p40. Ustekinumab was approved for psoriasis (2009), psoriatic arthritis (2013), Crohn’s disease (2016) and ulcerative colitis (2019). However, an important question is whether the promising effects of ustekinumab in IBD are the results of blocking IL-12, IL-23, or both cytokines. Antibodies exclusively targeting IL-23 via p19 are the focus of much research. Several anti-IL-23p19 antibodies, including risankizumab, brazikumab, mirikizumab, tildrakizumab and guselkumab, are in various stages of development to target IBD [35]. In 2017, guselkumab was approved by the FDA and EMA for psoriasis [43]. In 2018, tildrakizumab received US FDA, Australian Therapeutic Goods Administration and EMA approval for use in moderate to severe psoriasis [151]. The knowledge about the structure and composition of the cytokine:receptor complex, as well as signaling components, offers new possibilities for effective targeting strategies of both cytokines. Janus kinase inhibitors have been developed and evaluated in clinical trials. In 2018, tofacitinib was approved by the FDA and EMA for the treatment of ulcerative colitis [152]. Recently, a small-molecule inhibitor of Tyk2 blocked IL-23 signaling in vitro and inhibited disease progression in animal models of spondyloarthritis (SpA, [153]). Another example of therapeutics are proteins which interfere with cytokine:receptor binding. One example is an Alphabody, a protein scaffold featuring a single-chain antiparallel triple-helix coiled-coil fold, which sequesters site 3 (W156) on human IL-23 [154]. Blocking site 3 on p19 prevents its interaction with IL-23R and therefore IL-23 signaling. Additionally, small peptide antagonists to specifically target the IL-23R have been evaluated [155,156]. An highly efficient therapy might be a combination of different drugs targeting extracellular cytokine:receptor assembly and intracellular signaling pathways. Deep structural insights into cytokine:receptor binding provide new avenues for the development of novel therapeutic strategies.

## Figures and Tables

**Figure 1 cells-09-02184-f001:**
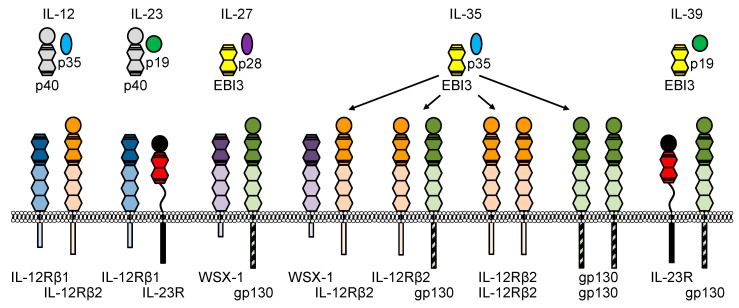
Cytokines and cognate receptors of the IL-12 family. Schematic overview of heterodimeric IL-12 type cytokines and their cognate receptors. IL-12, composed of IL-12p40 and IL-12p35, binds to the IL-12 receptor complex comprised of two receptors, IL-12Rβ1 and IL-12Rβ2. The heterodimeric cytokine IL-23 is formed by IL-12p40 and IL-23p19, and binds the IL-23 receptor complex comprised of IL-12Rβ1 and IL-23R. EBI3 and IL-27p28 form IL-27, which binds to WSX-1 and gp130, the signal transducer of IL-6. IL-35 is composed of EBI3 and IL-12p35. Four different IL-35 receptor complexes have been proposed: WSX-1/IL-12Rβ2, IL-12Rβ2/gp130, IL-12Rβ2/IL-12Rβ2 and gp130/gp130. The cytokine IL-39, formed by EBI3 and IL-23p19, binds to gp130 and IL-23R. Ig-domains are shown as circles and in the same color as cytokine-binding homology region (CHR). Membrane proximal fibronectin-type-III domains are presented in light colors. The WSXWS motifs within IL-12p40, EBI3 and the receptors are indicated by thick black lines and conserved cysteine residues by thin black lines.

**Figure 2 cells-09-02184-f002:**
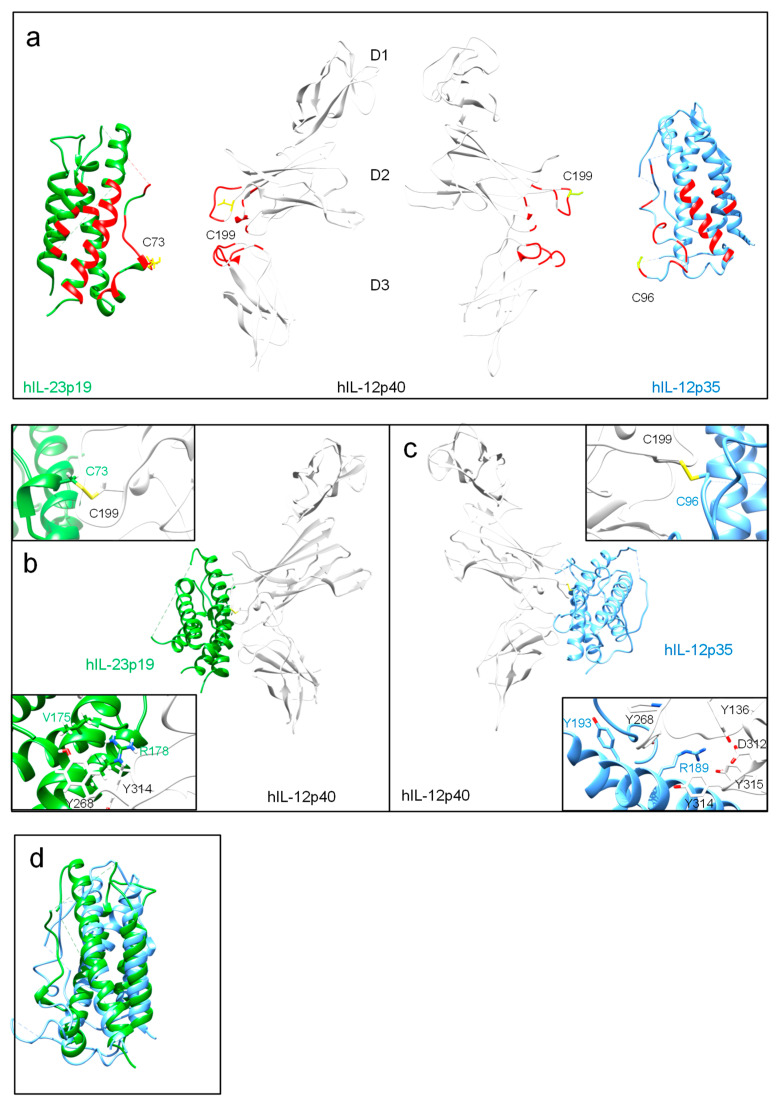
Structures of the heterodimeric cytokines IL-23 and IL-12. (**a**) Structure of hIL-23p19 (PDB: 5MXA), hIL-12p40 (PDB 5MXA), and hIL-12p35 (PDB: 3HMX). Location of site 1 residues are highlighted (red, yellow for cysteines). (**b**) Heterodimeric hIL-23. Close-up view on IL-23p19-IL-12p40 disulfide bond and hot spot amino acids for IL-12p40 interaction (inset). (**c**) Heterodimeric hIL-12. Close-up view on IL-12p35-IL-12p40 disulfide bond and hot spot amino acids for IL-12p40 interaction (inset). (**d**) Superpositioning of human IL-12p35 and IL-23p19. Molecular graphics images were produced using the UCSF Chimera package from the Resource for Biocomputing, Visualization, and Informatics at the University of California, San Francisco (supported by NIH P41 RR-01081).

**Figure 3 cells-09-02184-f003:**
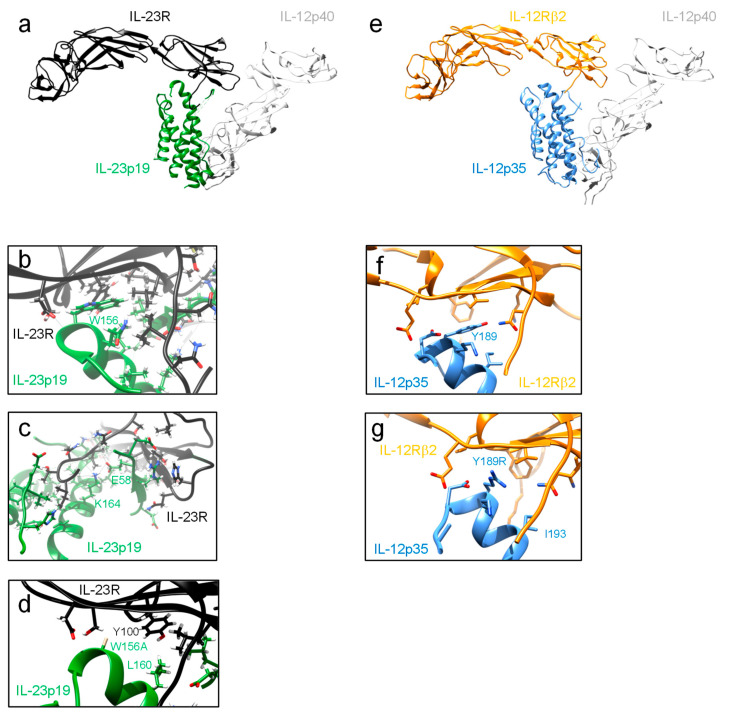
Structures of IL-23:IL-23R and IL-12:IL-12Rβ2. (**a**) Ribbon presentation of the hIL-23:hIL-23R structure (PSB 5MZV). (**b**) Close-up view of the hIL-23p19:hIL-23R interface surrounding the central W156. (**c**) Close-up view of the hIL-23p19:hIL-23R interface centering on the N-terminus of the hIL-23R and the IL-23p19 AB loop. (**d**) In silico mutation of W156 into A is displayed. Surrounding residues L160 (hIL-23p19) and Y100 (hIL-23R) are indicated. (**e**) Model of a complex of hIL-12Rβ2 and hIL-12 based on the crystal structures of the hIL-23:hIL-23R complex (PDB 5MZV) and the crystal structure of hIL-12 bound to an ustekinumab FAB (PDB 3HMX). (**f**) Model of hIL-12p35 (PDB 3HMX) bound to the hIL-12Rβ2. The conserved hIL-12p35 Y189 residue located at the top of helix D is highlighted within the site 3 interface. (**g**) Model of hIL-12p35 Y189R mutation. Residues I193 (hIL-12p35) and F99 and V100 (hIL-12Rβ2) corresponding to L160 (hIL-23p19) and Y100 (hIL-23R) are indicated. Molecular graphics images were produced using the UCSF Chimera package from the Resource for Biocomputing, Visualization, and Informatics at the University of California, San Francisco (supported by NIH P41 RR-01081).

**Figure 4 cells-09-02184-f004:**
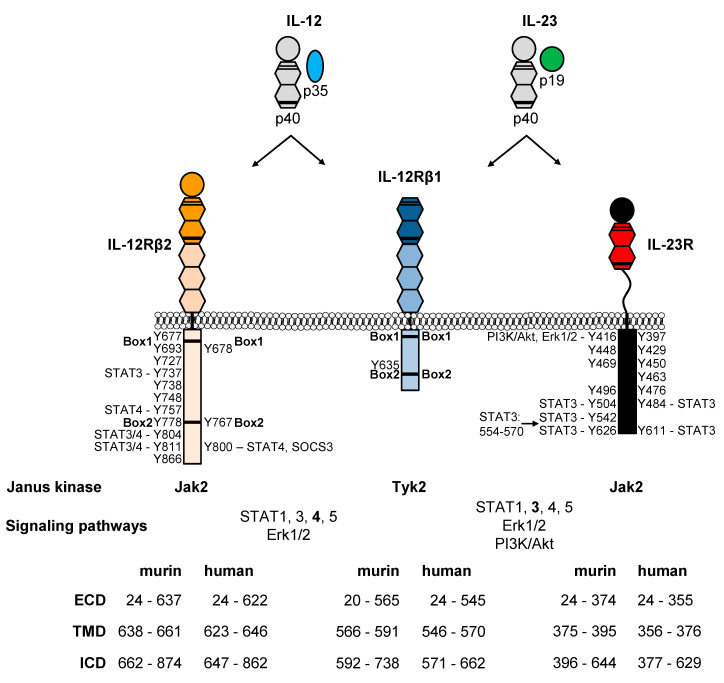
IL-12 and IL-23 receptor complexes and signal transduction. Schematic overview of IL-12 and IL-23 receptor complexes for mouse and human. Jak2 and Tyk2 are associated with the receptors. Box1 and Box2 motifs for Jak association and activation are highlighted. Tyrosine residues within the cytoplasmic domains are shown and STAT recruitment sites are highlighted. Intracellular signaling pathways induced by tyrosine-phosphorylation are indicated.

**Figure 5 cells-09-02184-f005:**
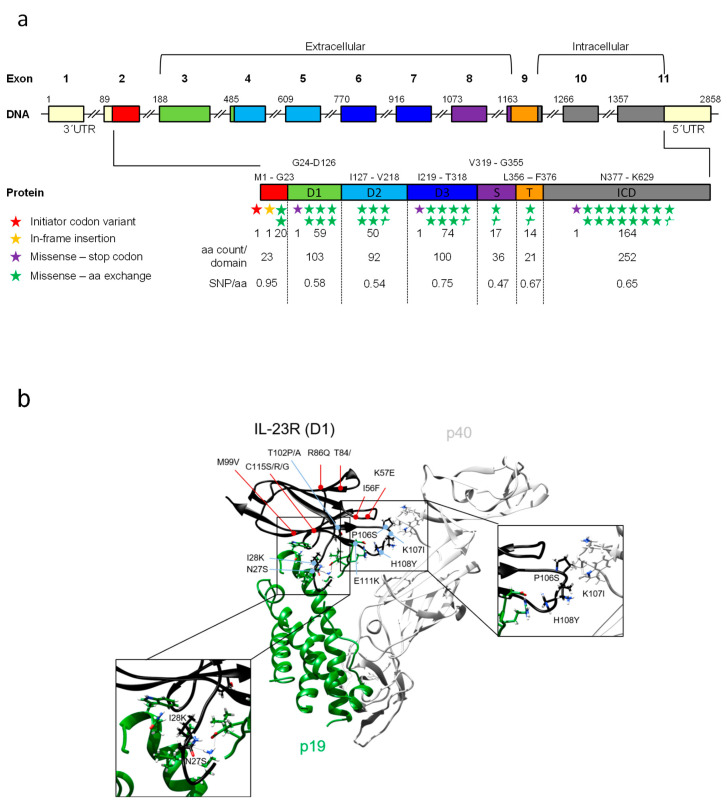
Single nucleotide polymorphisms of human IL-23R. (**a**) Schematic illustration of the intron-exon-structure and the corresponding translated human IL-23R protein. mRNA and amino acid numbers are given for exons/intron and domain borders, respectively (red: signal peptide; green: D1—domain 1; light blue: D2—domain 2; dark blue: D3—domain 3; purple: S—stalk region; orange: T—transmembrane domain; grey: ICD—intracellular domain). Moreover, number of missense mutations are included by colored stars (red star: initiator codon variant, yellow star: in-frame insertion, purple star: missense stop codon, green star: missense amino acid exchange). (**b**) Structure of hIL-23:hIL-23R D1 (PDB 5MZV). IL-23R SNPs unlikely to directly affect the site 3 interface are indicated in red, residues with potential deleterious effects are highlighted in blue. Sidechains of SNPs within site 3 interface and sidechain residues of IL-23p19 and IL-12p40 within 5 Å distance of these SNPs are displayed. Insets display close-up view of IL-23p19 and IL-12p40 interactions of the IL-23R receptor. Molecular graphics images were produced using the UCSF Chimera package from the Resource for Biocomputing, Visualization, and Informatics at the University of California, San Francisco (supported by NIH P41 RR-01081).

**Table 1 cells-09-02184-t001:** IL-12 Cytokine Family Members.

		α Subunits		
		p19	p28	p35
**β subunits**	**p40**	IL-23 [9]	IL-Y [7,8]	IL-12 [10,11,12,13,14]
	**EBI3**	IL-39 [15,16,17]	IL-27 [18]	IL-35 [19,20]

**Table 2 cells-09-02184-t002:** Similarities and Differences of IL-12 and IL-23 on Immune Response.

	IL-12	IL-23	Ref.
**transcription regulators**	T-bet, STAT4	RORγT, STAT3	[6]
**T cell differentiation**	TH1 cells	TH17 cells	[6]
**cytokine induction**	IFN-γ, TNFα	IL-17A, IL-17F, IL-22, TNFα	[35]
**inflammation**	pro-inflammatory cytokines, significant role of IL-12p40		[43]
**IBD (UC and CD)**	inflammation of gastrointestinal mucosa		[43]
**autoimmunity**	important role of IL-12p40		[44]
**psoriasis**	IL-12p40 as therapeutic target		[45]
**RA**	promotion of joint autoimmune inflammation	attenuation of joint autoimmune inflammation	[46]
**multiple sclerosis**	importance of multifunctional CD4^+^ T cells secreting IFN-γ, IL-17 and GM-CSF		[47]
**cancer**	antitumor activity	tumor-promoting and tumor suppressing effects	[37,48]
**infection**	important role in cell-mediated immunity against bacteria, fungi and intracellular protozoa	role in immunity to extracellular bacterial infections and fungal infections	[44]

IBD, inflammatory bowel disease; UC, ulcerative colitis; CD, Crohn’s disease; RA, rheumatoid arthritis.

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
