# Peer review of "IL-12 and IL-23—Close Relatives with Structural Homologies but Distinct Immunological Functions"

_cells, 2020, doi:10.3390/cells9102184_

Round 1

Reviewer 1 Report

Overall, well written review with nice illustrations.

it would be nice if the authors included a paragraph on the various immune response regulated by IL-12 and IL-23, highlighting the similarities and difference between the two. 

Author Response

Authors answer: We included Table 2 on page 3 to give an overview about similarities and differences of IL-12 and IL-23 on immune response. We also updated the references.

Reviewer 2 Report

A well organised and well written review article that also includes novel analysis of SNPs in the hIL-23R.

The article reviews the structural features of IL-12 and IL-23 and their receptors, a topic that has not been reviewed since an article was published by Vignal and Kuchroo in Nature Immunology in 2012 (DOI: 10.1038/ni.2366). It would perhaps have been interesting to give more emphasis to the topic of immunological function, as per the title of the review. However, it is acknowledged that this topic has been covered extensively in a number of other recent reviews including those by Hasegawa et al, Front. Immunol., 04, 2016 https://doi.org/10.3389/fimmu.2016.00479 and Wojno et al, Immunity, 2019  DOI: 10.1016/j.immuni.2019.03.011.

The authors discuss several anti-IL23 antibodies - however, tildrakizumab should also be included in the article. 

There are few typographical errors which should be corrected prior to publication. For example 'Graves opthalmopathy' should requires an apostrophe after the 's' in Graves' (lines 493 and  508). Check the script for appropriate and consistent use of italics.

Other minor errors were noted including, but not limited to, the following:

line 59 (gp130/gp130/,) should be gp130/gp130. 

line 92 - the sentence refers to a couple of reviews but then cites 6 papers

line 276 - constitutive should be changed to constitutively

line 370 - the sentence beginning 'Based on these studies,...' needs re-organising to make better grammatical sense.

line 380 - the sentence beginning 'Accordingly, IL23-R might...' needs re-organising to make better grammatical sense.

line 383 - 'expression of human' should be changed to 'expression by human'

line 399 - 31.233 SNPs should be 31,233 SNPs

line 427 - know should be corrected to 'now'

line 434 - the sentence beginning 'Therefore, to first identify...' needs re-organising to make better grammatical sense.

line 455 - references are missing

line 465 - correct spelling of 'esophagael'

line 479 - should read 'reticulum as an unfolded...' and 'surface levels and resulting in diminished cellular...'

line 483 - 'variant do' should read 'variant does..'

line 484 - 'a disease association was not studied...' should read 'a disease association has not been studied...'

line 491 - it is not clear how the 39 publications have been referenced here.

line 497 - 'show' should be 'shown'

line 522 - 'IL-12 has been' should be 'IL-12 was...'

line 533 - is the use of the word 'Accordingly' appropriate here?

line 545 - correct the spelling of therefore

line 546 - 'specific target' should read 'specifically target'

line 546 - IL-23R are evaluated should read 'IL-23R have been evaluated for'

Before publication the script needs to be carefully checked for gramatical errors and spelling mistakes.

Author Response

The article reviews the structural features of IL-12 and IL-23 and their receptors, a topic that has not been reviewed since an article was published by Vignal and Kuchroo in Nature Immunology in 2012 (DOI: 10.1038/ni.2366). It would perhaps have been interesting to give more emphasis to the topic of immunological function, as per the title of the review. However, it is acknowledged that this topic has been covered extensively in a number of other recent reviews including those by Hasegawa et al, Front. Immunol., 04, 2016 https://doi.org/10.3389/fimmu.2016.00479 and Wojno et al, Immunity, 2019  DOI: 10.1016/j.immuni.2019.03.011.

Authors answer: We included Table 2 on page 3 to give an overview about similarities and differences of IL-12 and IL-23 on immune response.

The authors discuss several anti-IL23 antibodies - however, tildrakizumab should also be included in the article. 

Authors answer: We included tildrakizumab in line 538 and added the following sentence (line: 550-551): In 2018, tildrakizumab received US FDA, Australian Therapeutic Goods Administration and EMA approval for use in moderate to severe psoriasis [146].

There are few typographical errors which should be corrected prior to publication. For example 'Graves opthalmopathy' should requires an apostrophe after the 's' in Graves' (lines 493 and  508). Check the script for appropriate and consistent use of italics.

Authors answer: We inserted an apostrophe after the “s” in Graves’ (lines 506 and 520/521). We checked the script and used italics for in vitro (line 171). We also corrected additional typographical errors (Table 1, lines 120, 148, 150, 152, 229, 239, 242, 253-255, 283, 310, 362, 390, 402, 423, 458, 459, 517, 522, 557 and 566). Figure 3 and 5 were changed because of typographical errors (3f and 3g). The figure legend for Figure 5 was updated.

Other minor errors were noted including, but not limited to, the following:

line 59 (gp130/gp130/,) should be gp130/gp130. 

Authors answer: We deleted “/” in line 59.

line 92 - the sentence refers to a couple of reviews but then cites 6 papers

Authors answer: We changed the sentence into: “….have been described in various reviews…” (line 93/93).

line 276 - constitutive should be changed to constitutively

Authors answer: We changed “constitutive” to “constitutively” (line 284).

line 370 - the sentence beginning 'Based on these studies,...' needs re-organising to make better grammatical sense.

Authors answer: We re-organized the sentence: “Accordingly, we speculate….” (line 378).

line 380 - the sentence beginning 'Accordingly, IL23-R might...' needs re-organising to make better grammatical sense.

Authors answer: We re-organized the sentence: “We and others showed that IL-23 induced sustained STAT3 activation despite increasing expression of SOCS3 indicating IL-23R might be no target of SOCS proteins [112,124,126]” (lines 388-389).

line 383 - 'expression of human' should be changed to 'expression by human'

Authors answer: We changed “of” to “by” (line 392).

line 399 - 31.233 SNPs should be 31,233 SNPs

Authors answer: We changed 31.233 to 31,233 (line 408).

line 427 - know should be corrected to 'now'

Authors answer: We changed “know” to “now” (line 436).

line 434 - the sentence beginning 'Therefore, to first identify...' needs re-organising to make better grammatical sense.

Authors answer: We re-organized the sentence: “An alternative experimental approach to manage the expanding SNP landscape is to identify missense mutations with altered biological function followed by subsequent analysis of disease association.” (lines 443-445).

line 455 - references are missing

Authors answer: The references are given in each paragraph of the respective SNP.

line 465 - correct spelling of 'esophagael'

Authors answer: We corrected the spelling of “esophageal” (line 477).

line 479 - should read 'reticulum as an unfolded...' and 'surface levels and resulting in diminished cellular...'

Authors answer: We corrected the sentence: “The G149R IL-23R variant is retained in the endoplasmic reticulum as an unfolded polypeptide, thereby reducing cell surface levels and resulting in diminished cellular activation by IL-23 [140]).” (line 491)

line 483 - 'variant do' should read 'variant does..'

Authors answer: We corrected “do” to “does” (line 495).

line 484 - 'a disease association was not studied...' should read 'a disease association has not been studied...'

Authors answer: We wrote: “…A disease association was has not been studied to date.” (line 496).

line 491 - it is not clear how the 39 publications have been referenced here.

Authors answer: We inserted (dbSNP) (line 503) to make clear that these references are listed in the SNP data bank.

line 497 - 'show' should be 'shown'

Authors answer: We wrote: “shown”…(line 509).

line 522 - 'IL-12 has been' should be 'IL-12 was...'

Authors answer: We wrote: “IL-12 was”…(line 535).

line 533 - is the use of the word 'Accordingly' appropriate here?

Authors answer: We deleted “Accordingly,” (line 546/547).

line 545 - correct the spelling of therefore

Authors answer: We wrote: “therefore”…(line 560).

line 546 - 'specific target' should read 'specifically target'

Authors answer: We wrote: “specifically target”…(line 561).

line 546 - IL-23R are evaluated should read 'IL-23R have been evaluated for'

 Authors answer: We wrote: “IL-23R have been evaluated for”…(line 561).

Before publication the script needs to be carefully checked for grammatical errors and spelling mistakes.

Authors answer: We corrected the following grammatical errors and spelling mistakes: Table 1, lines 120, 148, 150, 152, 229, 239, 242, 253-255, 283, 310, 362, 390, 402, 423, 458, 459, 517, 522, 557 and 566.